# Effects of Arbuscular Mycorrhization on Fruit Quality in Industrialized Tomato Production

**DOI:** 10.3390/ijms21197029

**Published:** 2020-09-24

**Authors:** Ramona Schubert, Stephanie Werner, Hillary Cirka, Philipp Rödel, Yudelsy Tandron Moya, Hans-Peter Mock, Imke Hutter, Gotthard Kunze, Bettina Hause

**Affiliations:** 1Leibniz Institute of Plant Biochemistry, Department of Cell and Metabolic Biology, 06120 Halle, Germany; ramona.schubert@ipb-halle.de (R.S.); stephanie.werner@julius-kuehn.de (S.W.); 2Julius Kühn-Institute (JKI), Federal Research Centre for Cultivated Plants, Institute for Biosafety in Plant Biotechnology, 06484 Quedlinburg, Germany; 3INOQ GmbH, 29465 Schnega, Germany; cirka@inoq.de (H.C.); philipproedel@gmx.de (P.R.); hutter@inoq.de (I.H.); 4Leibniz Institute of Plant Genetics and Crop Plant Research (IPK), 06466 Gatersleben, Germany; moya@ipk-gatersleben.de (Y.T.M.); mock@ipk-gatersleben.de (H.-P.M.); kunzeg@ipk-gatersleben.de (G.K.)

**Keywords:** arbuscular mycorrhiza, BRIX value, carotenoids, free amino acids, fruit quality, hydroponic cultivation, phosphate, tomato, transcript profiling (RNAseq), yield

## Abstract

Industrialized tomato production faces a decrease in flavors and nutritional value due to conventional breeding. Moreover, tomato production heavily relies on nitrogen and phosphate fertilization. Phosphate uptake and improvement of fruit quality by arbuscular mycorrhizal (AM) fungi are well-studied. We addressed the question of whether commercially used tomato cultivars grown in a hydroponic system can be mycorrhizal, leading to improved fruit quality. Tomato plants inoculated with *Rhizophagus irregularis* were grown under different phosphate concentrations and in substrates used in industrial tomato production. Changes in fruit gene expression and metabolite levels were checked by RNAseq and metabolite determination, respectively. The tests revealed that reduction of phosphate to 80% and use of mixed substrate allow AM establishment without affecting yield. By comparing green fruits from non-mycorrhizal and mycorrhizal plants, differentially expressed genes (DEGs) were found to possibly be involved in processes regulating fruit maturation and nutrition. Red fruits from mycorrhizal plants showed a trend of higher BRIX values and increased levels of carotenoids in comparison to those from non-mycorrhizal plants. Free amino acids exhibited up to four times higher levels in red fruits due to AM, showing the potential of mycorrhization to increase the nutritional value of tomatoes in industrialized production.

## 1. Introduction

Tomato (*Solanum lycopersicum*) is an important plant for human nutrition, especially for the so-called low-fat Mediterranean diet. Economically it is the most important fleshy fruit vegetable worldwide with a production yielding 177 million t/year and a per capita consumption of 18 kg (FAO 2018). Tomato production is commercialized for a wide range of products. Tomatoes are freshly marketed, but also sold as conserves and processed into juices. These diversified downstream products correspond to different cultivation methods, such as cultivation in greenhouses and in open fields, respectively. Tomatoes for fresh markets are mainly grown in greenhouses to enable a continuous harvest. However, due to conventional breeding the fruit has faced a decrease in flavor and nutritional value over the last centuries [1]. Additionally, high tomato production relies on high amounts of nitrogen and phosphate fertilizers; however, rock phosphate is a limited resource that may be depleted 70–40 years from now [2]. Taken together with the growing human population with a demand for more and nutrient rich food, strategies are needed to enhance phosphate absorption by the plant and improve fruit nutritional values.

In the last few decades, we gained greater awareness of the positive effects of root symbionts such as arbuscular mycorrhizal fungi (AMF) on plant performance. Arbuscular mycorrhiza (AM) describes a mutualistic symbiosis between a plant and a fungus of the phylum Glomeromycotina (formerly Glomeromycota [3]. Presumably, this 450 million-year-old interaction allowed the evolution of plants on land due to the more efficient phosphate uptake from the rhizosphere through the AMF [4]. As a result of a complex molecular dialogue with its host plant, the fungus colonizes the root cortexes of the plants, develops intercellular hyphae and forms highly branched structures within the cortex cells, the arbuscules [5]. Both partners benefit from this interaction—the fungal partner helps the plant to absorb nutrients from the soil, in particular phosphorus and nitrogen, through its expanded mycelium [6,7]. Depending on the species, the complete plant phosphate can be delivered via the mycorrhizal pathway, reducing the direct root pathway [8]. In return, the plant supplies the obligatory biotrophic fungus with carbohydrates, which are taken up by the fungus as hexoses and fatty acids [9]. The advantages for the plant resulting from the interaction with an AMF are, however, not only an improved nutrient supply but also increased resistance against pathogens, and improved tolerance to water and salt stress and to pollutants [10,11,12]. Other well-described positive effects of AM are an early onset of flowering and an increased yield and number of marketable fruits [13,14,15,16,17]. As a consequence, AMF are more and more integrated into plant production systems, as they contribute to a reduction of chemical fertilizers and pesticides. and are therefore potentially important components of sustainable food production [18]. In this way, microbial inoculants containing AMF as “biostimulants” might provide one solution to face the above-mentioned future challenges in human nutrition [19].

For the tomato, however, not all of the studies showed an increase in fruit yield as a response to root colonization [13,20], but mycorrhization shortens the vegetative growth period of tomato plants and thus leads to an earlier maturation of flowers and fruits [21]. In addition, there is evidence that the interaction of tomato plants with AMF increases the nutrient quality by increasing the contents of citric acid, carotenoids and certain amino acids and the antioxidant capacity of the fruits [13,22,23,24]. Furthermore, tomatoes from mycorrhizal plants possess higher anti-estrogenic properties, do not show genotoxic effects [23] and do not represent an increased allergenic potential for humans [25].

Still, this outcome was dependent on the tomato cultivar and fungal species used. Besides that, most of the studies were conducted in an experimental setup under laboratory conditions and therefore far from industrial conditions. Many growers of greenhouse tomatoes prefer a “soil-less” production due to high yields and good quality of the fruit. These soil-less, hydroponic cultivation systems provide better control of fertilization and irrigation. It is not known, however, whether tomato plants grown under these conditions are able to interact with AMF to lead to enhanced quality of the fruit.

Therefore, our study aimed to discover growth conditions suitable for tomato producers and AM formation in the greenhouse. Commercially used tomato cultivars (cv. Brioso and Picolino, the latter grafted on cv. Maxifour) were chosen in combination with the model AM fungal species *Rhizophagus irregularis*, which is distributed worldwide and able to adapt to foreign plant species [26]. Moreover, its genome has been sequenced [27], and *R. irregularis* is easy to cultivate and to maintain in trap cultures. At the same time, genes should be identified that are differentially expressed between fruits from non-mycorrhizal and mycorrhizal plants in order to find markers indicating tomato quality improvement. The study was complemented by targeted analysis of metabolites and minerals in fruits to find effects of mycorrhiza on the quality of tomatoes grown under producer conditions.

## 2. Results

### 2.1. AM Establishment in Hydroponics Is Influenced by Growth Substrate and Fertilization

AM mostly establish themselves under abiotic stress conditions of the plant, mainly limitations of phosphate availability. To achieve AM formation under tomato cultivation conditions in the greenhouse, fertilizer composition and suitability of the commercially used substrate need to be adjusted.

Therefore, plants of the commercially grown tomato cultivar Brioso inoculated with *R. irregularis* were grown in expanded clay as a nutrient free substrate to define a phosphate (Pi) concentration efficient for normal plant growth but allowing AM formation. Fertilizer containing 2.7, 6.7 or 10.7 mM Pi was applied. Under 10.7 mM Pi, plant growth was similar to control plants fertilized with 13.3 mM Pi and remaining non-inoculated, and a mycorrhization was achieved exhibiting a rate of around 35% (Figure 1a). In addition, the effect of reduced Pi supply was tested for its effect on yield. Brioso plants grown under industrial tomato-producing conditions were either fertilized with 13.3 mM or 10 mM of Pi. Yield in terms of totally harvested tomatoes and number of tomatoes per plant was recorded and did not show significant differences (Figure 1b).

Therefore, a fertilizer containing 10.7 mM Pi was used to test AM establishment in substrates used by tomato producers. Among these substrates, coco mats are widely used in hydroponic (soil-less) tomato cultures in greenhouse. To test this substrate, tomato plants cv. MicroTom were used as a fast growing and easy to handle cultivar. While growing MicroTom plants in expanded clay mixed with crushed coco mat in a ratio of 1:1, no mycorrhization was detectable five and eight weeks after inoculation. At the same time, plants grown and inoculated in pure expanded clay reached a colonization rate of 90%. This result showed that coco mats are not suitable for AM formation. Therefore, a second substrate used by tomato producers (“Hawita special mix”) and consisting of 80% coco peat, 10% clay, 10% peat and 3 kg lime per cubic meter was tested. In this substrate, MicroTom plants did not show root colonization after five weeks of inoculation, but reached a rate of 30% of mycorrhization after eight weeks.

Finally, the substrate “Hawita special mix” and fertilizer containing 10 mM Pi were used to grow tomato plants cv. Picolino (grafted on cv. MaxiFour) inoculated or not with *R. irregularis* in industry-like conditions in the greenhouse of INOQ GmbH (Figure 2a). Grafted plants are commonly used in tomato production to combine yield stability with tolerance of roots against biotic and abiotic stresses. Fruits of these plants were used for all following experiments. Root colonization was tested five, nine, 13 and 17 weeks after inoculation. The tests revealed that AM formation started after nine weeks and was well established after 17 weeks after inoculation (Figure 2b–g).

These data show unequivocally that AM formation using *R. irregularis* in the greenhouse under tomato producer conditions can be achieved, although it is delayed in comparison to plants grown under laboratory conditions using nutrient-poor substrates. Due to the long growth period, however, the mycorrhization rates achieved were relatively high and reached levels also known from plants grown under stronger P-limitations and laboratory conditions (e.g., see [24]).

### 2.2. Mycorrhization of Plants Resulted in Minor Effects on Tomato Fruit Transcriptome

Previously published reports describe that mycorrhization led to increased yield in tomato grown under field conditions [13,28]. Using the set-up described above, there were, however, no effects by mycorrhization on the total yield of tomatoes grown in the greenhouse. Total yield of plants was exactly the same in inoculated plants vs. non-inoculated plants (Appendix A).

To detect effects of AM on tomato fruit quality, green and red fruits were harvested when mycorrhization was well established (17 weeks after inoculation). For getting first insights on putative physiological effects on red fruits, the transcriptomes of green and red fruits were determined. The RNAseq approach resulted in 38.2 to 60.8 million mapped reads per sample. To visualize an effect of mycorrhization on gene expression, hierarchical clustering of all differentially expressed genes between green and red fruits of mycorrhizal and non-mycorrhizal plants was performed. The heatmap showed two distinct clusters between green and red fruits, indicating clear differences in gene expression between the two developmental stages (Figure 3). In contrast, sub-clusters within the green or red stages indicative of effects of mycorrhization were not visible. This led to the conclusion that mycorrhization rarely affected gene expression in fruits.

The comparison of the numbers of differentially expressed genes (DEGs) in single comparisons supported this conclusion: 13,003 DEGs were found between green and red fruits, whereas when comparing fruits from mycorrhizal and non-mycorrhizal plants, only 80 and 51 DEGs were identified from green and red fruits, respectively (Appendix A). The fold changes in expression of all DEGs in red fruits were below |2|. Additionally, differences in gene expression in the red stage might be too late to affect the metabolome of the red fruits. Therefore, we focused on DEGs in green fruits. Except two, all of these genes were upregulated in mycorrhizal plants. A subset of 15 genes was analyzed via qPCR to validate the RNAseq data. The results showed the same tendency with higher expression of the selected genes (except lower for *ACC-oxidase5*) in green fruits from mycorrhizal plants compared to green fruits from non-mycorrhizal plants (Table 1). However, due to high variances partially caused by a very low expression under non-mycorrhizal conditions, these differences were not significant.

Overall, the data indicated an only weak effect of arbuscular mycorrhization on the fruit transcriptome in tomato under greenhouse conditions.

### 2.3. AM Effect on Tomato Fruit Quality

Despite the low effects of AM on the tomato fruit transcriptome, putative increases of health or taste-promoting compounds in red fruits have been tested.

Fruit sweetness strongly influences fruit taste. An easy indication of fruit sweetness is the determination of the BRIX value representing the content of dissolved solids, mainly sucrose. The analysis showed by trend higher BRIX values of red fruits from mycorrhizal plants (Figure 4). Therefore, the data suggest a positive effect of mycorrhization on tomato taste in terms of increased sweetness.

In addition, the levels of minerals, carotenoids and amino acids were determined from green and red fruits harvested from non-mycorrhizal and mycorrhizal plants (Appendix A, Table 2, Figure 5). Quantitative analyses of minerals revealed no differences in green and red fruits from non-mycorrhizal and mycorrhizal plants (Appendix A). Regarding carotenoids, the contents of lutein, zeaxanthin, lycopene and β-carotene showed clear differences between the two developmental stages, but no significant differences between fruits of the same developmental state from plants that were mycorrhizal or not. Nevertheless, a tendency was visible in red fruits for an increase in the levels of lycopene and β-carotene due to mycorrhization (Table 2).

With respect to free amino acids, green fruits from non-mycorrhizal and mycorrhizal plants did not show significant differences in their levels of 19 amino acids, which were analyzed (Figure 5a). Analysis of red fruits, however, revealed a significant increase in the levels of all amino acids except proline due to the mycorrhization of plants (Figure 5b). Among the amino acids showing increased abundance in the fruit from mycorrhizal plants, glutamine, glutamate, asparagine and phenylalanine were the most responsive amino acids with an increase up to four-fold. These data show that mycorrhization of tomato plants resulted in an increase in carotenoids and amino acids in fruits, thereby leading to higher potential for human health and nutrition.

## 3. Discussion

In this study we showed that it is possible to establish a mycorrhizal interaction in a hydroponic cultivation system, which is frequently used in greenhouses by producers of fresh tomatoes. Successful colonization of plants by an AMF led to an improvement of fruit quality, as shown by an increase of the content of sugars (increased BRIX values), carotenoids and amino acids.

It has been shown previously that interaction of tomato plants with AMF may result in improved fruit production in terms of increased yield and improved fruit quality [13,22,23,24]. Most of those studies were performed, however, on plants grown under laboratory or open field conditions, where plants were grown under phosphate limitation to increase mycorrhization. Moreover, in those experiments, mainly quartz sand was used as substrate, which is not suitable for hydroponics [13,22]. The use of hydroponic systems enables tomato growers to control the growing medium and to maintain proper nutrient conditions, resulting in high yields and good fruit quality. These nutrient conditions might, however, prevent mycorrhization, since high nutrient supply, e.g., high level of Pi, restricts the interaction of plants with AMF [29,30,31,32]. High Pi levels trigger a complex anti-symbiotic syndrome, which results in a strong repression of root colonization by AMF [33,34]. Pre-experiments shown here revealed that the use of a fertilizer with slightly reduced phosphate content (75% of “normal” Pi content) did not affect the yield, but enabled a colonization of tomato roots by *R. irregularis*. The mycorrhization rate obtained was lower than under stronger Pi-deficiency, but appeared to be sufficient to test the effects of mycorrhization on fruits. However, this set-up enabled an analysis of the effect of an AM on the tomato fruit quality without affecting the nutrition status of the plants.

Further, the substrate identified to enable a successful mycorrhization was a mixture of coco peat, clay and peat. Pure coco mats, widely used in commercial production of tomatoes in greenhouses, prevented mycorrhization of tomato plants, even if crushed coco mats were mixed with other substrates, such as expanded clay. It is not known which components in coco mats are responsible for this effect. Pure coco mats might have some anti-fungal properties, which may affect the AMF directly. In addition, this substrate might have high water percolation properties such that hyphae cannot attach to the root.

Although a mycorrhization was achieved using Pi-reduced fertilizer and the special substrate mix, it took several weeks to detect a successful mycorrhization. That relatively long time and the relatively low levels of colonization could depend on several factors. On the one hand, under constant fertilization by drip irrigation, levels of both N and P might be still too high and negatively affect the symbiotic interaction [35]. On the other hand, fruit production itself is a major sink for carbon, and therefore might limit the available carbohydrates for the fungal partner leading to a decreased colonization [13]. Even the type of rootstock used for grafting might influence the capability for mycorrhization. To address these issues, other substrates and fertilizers, but also different tomato cultivars and AMF with higher capabilities to colonize tomato plants might be usable to improve mycorrhization. This will need further tests, e.g., with a mix of different AM fungal species.

The tomato is among the plant species which respond well to AM symbiosis in terms of systemic effects on fruit features [36,37]. Using the system described above, we did not find, however, effects of mycorrhization on the yield of tomatoes from cv. Picolino/Maxifour. This contrasts with previously published studies, which showed a clear increase of yield from mycorrhizal tomato plants in comparison to non-mycorrhizal plants [13,22,24,38]. It was assumed that interaction with AMF can affect tomato reproductive growth via several effects, such as acceleration of flowering and fruit development [24], and increases in the total number of flowers [28], the number of flowers per truss and the proportion of flowers setting fruits [39]. It is, however, necessary to mention that most of those studies used an extreme reduction in fertilization with phosphate, thereby enabling a very strong and effective interaction between plant and AMF. This would contrast our attempts to prevent reductions in total yield, which is a main parameter for tomato producers. Moreover, the use of the tomato cultivar and the fungal species may determine whether increase in yield can be obtained. Here, former studies used model (laboratory-used) cultivars of tomato, such as cv. MicroTom [24] and M19 [38], and other fungal species, such as *Funnelliformis mossea* [22,24] and *Glomus fasciculatum* [38].

The slight reduction of Pi accompanied by a rather low and late mycorrhization is also reflected in only minor differences in the transcriptome of mycorrhizal vs. non-mycorrhizal plants. Only 80 and 51 DEGs were identified from green and red fruits, respectively. This is in contrast to the number of DEGs identified by Zouari et al. (2014), who described more than 700 DEGs in fruits at the turning ripening stage from mycorrhizal and non-mycorrhizal plants [40]. Moreover, there was no overlap in the DEGs identified here with those identified by Zouari et al. (2014). Nevertheless, among the DEGs identified in green fruits from cv. Picolino/Maxifour, we found genes, which might be involved in processes regulating fruit maturation (e.g., *DEHYDRIN1*, a desiccation-related gene and *ERF13)*. Most importantly, genes encoding proteins which might be related to fruit nutrition were also differentially expressed and showed increased transcript levels in green fruits from mycorrhizal plants. Among these genes, we found genes encoding malic enzyme, 2S albumin seed storage protein, vicilin and an aminotransferase.

Although these changes in green fruits appear to be only minor, changes in metabolite levels of red fruits were obvious. Fruits from mycorrhizal plants exhibited a tendency to have higher BRIX values and higher contents of lycopene, and a significant increase in free amino acids in comparison to fruits from non-mycorrhizal plants. The mycorrhiza-induced increase in lycopene was also found in other studies [22,23], although molecular mechanisms leading to enhanced carotenoid levels have not been identified yet [22]. Increased lycopene or β-carotene levels indicate, however, a more general effect of AM on health properties of tomato fruits. Lycopene, as major carotenoid in red tomato fruits [41], has a high antioxidant capacity and might act protectively on cardiovascular risk factors, although the efficacy is not unequivocally demonstrated and showed high variability [42].

Another important finding of our investigation originates from the quantitative analysis of free amino acids in green and red fruits from non-mycorrhizal and mycorrhizal plants. In comparison to red fruits from non-mycorrhizal plants, red fruits from the mycorrhizal plants had significantly higher levels in all amino acids, except proline. The most obvious changes were found for glutamine, glutamate, asparagine and phenylalanine. Higher contents of glutamine and asparagine have also been found in fruits of tomato plants interacting with *F. mosseae*, but only in the turning stage [24]. These data might support the general benefit of an application of AM to improve the nutritional value of tomatoes produced not only in field conditions, but also in hydroponic systems in greenhouses.

In summary, mycorrhization of commercially used tomato cultivars grown in hydroponic systems affects fruit quality, but not yield, at least for the tomato cultivar and mycorrhiza strain used in this study. This is unlike other species, such as the strawberry [43], cucumber [44] and cotton [14], which showed positive effects of AM on yield and fruit quality. Moreover, specific effects on the tomato fruit might depend on the tomato cultivar, the cultivation condition and the AMF.

## 4. Materials and Methods

### 4.1. Plant Material and Growth Conditions

Plants of *Solanum lycopersicum* cv. Brioso (Wittenberg Gemüse GmbH, Wittenberg, Germany) and cv. MicroTom were grown in expanded clay of 2–5 mm particle size (Original Lamstedt Ton, Fibo ExClay, Lamstedt, Germany) in 21 and 12 cm pots from May until August in 2017 and from February until April in 2019, respectively, in the IPB greenhouse (Leibniz Institute of Plant Biochemistry, Halle/Saale, Germany) with no additional light. For substrate tests, expanded clay was mixed with crushed coco mates (CANNA Continental, Los Angeles, CA, USA) in 1:1 ratio or completely substituted by Hawita special mix containing 10% clay, 10% peat, 80% coco peat and 3 kg lime per cubic meter (HAWITA Gruppe GmbH, Vechta, Germany). Plants were watered with Aqua dest. and fertilized with 10 mL Long Ashton fertilizer [45] containing 2.7, 6.7 and 10.7 mM phosphate twice a week.

Inoculation with *Rhizophagus irregularis* strain K8/QS69 was done with one-week-old seedlings and cuttings of cv. MicroTom and cv. Brioso, respectively. Enrichment of the fungus by previous co-cultivation with leek (*Allium porrum* cv. Elefant) and inoculation of tomato plants was done as described [46].

At INOQ GmbH, wild type *Solanum lycopersicum* cv. Picolino grafted on cv. MaxiFour (Jahnataler Jungpflanzen Tänzler GbR, Hof/Sachsen, Germany) grown for 4 weeks in perlite blocks (2 plants per block) were placed on top of Hawita special mix in the greenhouse. A plastic foil separated the perlite block from the hydroponic substrate for better rooting. Inoculation was carried out with 1 g/L substrate of the commercial mycorrhiza inoculum INOQ Advantage, a powder with highly condensed mycorrhiza propagules (90.000/g). After six weeks the plastic foil was removed to let plants root into the hydroponic substrate.

A drip irrigation system was installed to ensure a constant fertilization. As fertilizer a mix of different nutrients was applied as follows: 1 mL/L YaraTera Calsal (YARA GmbH, Dülmen, Germany) 0.5 g/L Yaraliva Calnit (YARA GmbH), 1 g/L YaraTera Kristalon Weismarke (YARA GmbH), 0.015 g/L iron chelate, 0.25 g/L Epsom salt. This composition was used during early vegetation period from February to May with three irrigation cycles in February and up to 30 from May on. To adjust fertilizer input to hot summer season Yaratera Kristalon Weismarke, iron chelate and Epsom salt were reduced tenfold. The number of irrigation cycles was controlled by the extent of drainage as hydroponic substrate should show an overall drainage of 30%. Electric conductivity of drain was measured to ensure correct water and fertilizer input for optimal plant growth especially during the hot summer season.

For consistent pollination, a set of three bumblebee boxes (Katz Biotech AG, Baruth, Germany) was released during the period of tomato growth. Optimal plant growth was ensured by hanging the plants on hooks and regular removing of adventitious shoots, 2–3 times per week. Leaves were removed to let fruits ripen, with an equilibrium of a maximum of 3 leaves per week. As soon as a panicle of tomatoes ripened, the 1–2 shoots above it were removed. Only 1 panicle should ripen at a time. With continuous growth of the main stems plants were hanged further along the ceiling to allow stems to get as long as five meter with permanent blossoming and fruit ripening.

### 4.2. Determination of Mycorrhization Rate

Around 2 cm of the middle part of the root system was harvested at the indicated time points and stained with black ink (Sheaffer Skrip jet black, Sheaffer Manufacturing, Madison, WI, USA) according to [47]. The colonization rate was determined using a stereomicroscope via gridline-intersection method [48] or according to [49].

### 4.3. Transcriptomics and Quantitative Real-Time PCR (qPCR)

Mature green and red fruits of eight Picolino/Maxifour plants either mycorrhizal or non-mycorrhizal were harvested at INOQ GmbH, opened to remove the seeds and the remaining pericarps immediately frozen in liquid nitrogen. Pericarps were lyophilized (Christ Gefriertrocknungsanlagen GmbH, Osterode, Germany), homogenized and pooled to unify two fruits in one sample resulting in 16 samples. RNA was isolated using the RNeasy Plant Mini Kit (Qiagen, Hilden, Germany) with eluting only once in 30 µL water. DNA was removed using DNA-free^TM^ Kit (Thermo Scientific, Waltham, MA, USA) according routine DNase treatment and rigorous DNase treatment including 1:1 RNA dilution and 2 µL DNase for red and green fruit samples, respectively. RNA integrity was checked using Agilent Bioanalyzer 2100 (Agilent Technologies, Santa Clara, CA, USA) showing RIN factors above 8. Minimum 1 µg RNA was sent for Ilumina sequencing producing paired-end reads of 150 bp at Novogene UK Company Limited. Bioinformatics were performed by Novogene using the DESeq2 R package with read counts as input [50] and the Benjamini–Hochberg approach for p-value adjustment for DEG analysis. Hierarchical clustering analysis was carried out using log10 (FPKM+1) values of DEGs within all comparison groups. Sequence data (FASTQ files from RNA-seq experiment) can be found in the ArrayExpress database under accession number E-MTAB-9419.

To validate the RNAseq approach, 1 µg of the RNA samples of green fruits was used for cDNA syntheses using ProtoScript II First Strand cDNA Synthesis Kit (New England BioLabs GmbH, Frankfurt/Main, Germany) with oligo(dT) primers. cDNA samples were diluted 1:20 each and 3 µL used for qPCR reaction using EvaGreen QPCR Mix II without ROX (Bio & Sell, Feucht/Nürnberg, Germany) and gene specific primers (see Appendix A). qPCR was performed in the CFX Connect cycler (Bio-Rad Laboratories, Hercules, CA, USA) according to following protocol: 15 min at 95 °C, followed by 40 cycles 15 s at 95 °C and 30 s at 56 °C followed by melt curve generation via fluorescence detection between 60 °C and 95 °C for 1 s every 0.5 °C. Expression of all genes of interest was calculated relative to *TIP41*-expression according the ΔCq-method [51].

### 4.4. Determination of BRIX Values

One milliliter of water was added to 100 mg of non-pooled, lyophilized material of red fruits of mycorrhizal and non-mycorrhizal plants (cv. Picolino/Maxifour). After careful mixing, samples were incubated for three hours at room temperature to dissolve the soluble solids followed by a centrifugation step for 1 min at 7000× *g*. The resulting supernatant (juice) was applied to a refractometer and the BRIX value determined.

### 4.5. Determination of Amino Acids, Carotenoids and Minerals

Freeze-dried samples (20 mg) were extracted in 400 µL methanol with shaking at 70 °C for 30 min. After adding 200 µL CHCl_3_ and 400 µL pure water (18.2 MΩ cm; Merck Millipore, Darmstadt, Germany), the samples were centrifuged (18,000× *g*, 10 min, 4 °C), and an 80 µL aliquot of the aqueous methanol phase was vacuum-dried and re-dissolved in 10 µL water. The solute was derivatized using an AccQ Fluor kit (Waters, Eschborn, Germany), following the manufacturer’s instructions. A serially diluted standard mixture of amino acids was derivatized similarly for quantification purposes. A 1 µL aliquot was separated by an AccQ-Tag Ultra 1.7 µm, 2.1 × 100 mm UPLC column (Waters), as recommended by the manufacturer. Amino acids were detected by excitation at 266 nm using an emission wavelength of 473 nm, and identified and quantified by comparison with authentic amino acid standards with the help of the software Empower (Waters).

Carotenoids were determined by HPLC with UV detection (Waters UPLC with diode array detector; Eschborn, Germany). Samples (150 mg) were extracted with 600 µL of acetone containing 10 µM KOH. After shaking for 10 min at RT, samples were centrifuged and the pellet re-extracted twice with each 500 µL of the extraction solvent. The final extract was centrifuged again. HPLC separation was performed on a C30 column (150 × 4.5 mm) with a precolumn of the same material (YMC, Dinslaken, Germany). Solvents used were methanol (eluent A), 80% methanol with 0.2% (*w*/*v*) ammonia acetate (eluent B) and tert. butylmethylether (eluent C). The flow rate was 1.0 mL min^−1^ and the initial solvent composition was 85% A, 5% B and 10% C which was kept for 3 min after injection. In the following two minutes, the composition changed to 75% A, 5% B and 20% C. The gradient then changed to 27% A, 5% B and 68% C until min 18. Within one further minute, the final conditions were reached with 15% A, 5% B and 80% C. Re-equilibration was run for 5 min prior to injection of the next sample.

For mineral analysis, plant material was weighed into PTFE digestion tubes and concentrated nitric acid (0.5 mL; 67–69%, Bernd Kraft, Duisburg, Germany) was added to each tube. After 4 h of incubation, samples were digested under pressure using a high-performance microwave reactor (Ultraclave 4; MLS, Leutkirch, Germany). Digested samples were transferred to Greiner centrifuge tubes and diluted with de-ionized water to a final volume of 8 mL. Elemental analysis was carried out using inductively coupled plasma–mass spectrometry technique (ICP-MS, Sector Field High Resolution (HR)-ICP-MS, ELEMENT 2, Thermo Fisher Scientific, Dreieich, Germany) with Software version 3.1.2.242. For sample introduction a SC-2 DX Autosampler (ESI, Elemental Scientific, Mainz, Germany) was used. A six points external calibration curve was set from a certified multiple standards solution (Bernd Kraft, Duisburg, Germany). The standard reference material Tomato Leaves (NIST, 1573a) was used to verify data precision and accuracy (data not shown). The elements Rodium (Rh) and Germanium (Ge) were infused online and used as internal standards for matrix correction.

### 4.6. Statistical Analyses

All data were analyzed by one-factorial analysis of variance (ANOVA) and Bonferroni correction was included for multiple data sets. Tukey’s HSD post hoc test served as a multiple comparison test. Differences were considered significant at a probability level of *p* < 0.05. All statistical analyses were performed using the R software (R Core Team; https://www.r-project.org/).

## 5. Conclusions

We showed that the use of arbuscular mycorrhizal fungi as biostimulants is possible, even in a hydroponic set-up as commonly used in current industrial in-door tomato production systems. Prerequisites taken into account were (i) the use of a substrate which contained only low amounts of coco peat, and (ii) the adjustment of fertilization with phosphate until a reduced level was reached that allowed the establishment of the symbiosis without impairing plant growth and fruit yield. Here, an N:P ratio of minimum 10 allowed a mycorrhization in a commercial hydroponic system. As a result, mycorrhization of hydroponically grown tomato plants led to an increase in sugar content, as shown by increased BRIX values in lycopene, and in the levels of free amino acids. With this, the use of mycorrhiza enhances the nutritional quality of tomatoes commercially grown in greenhouses.

## Figures and Tables

**Figure 1 ijms-21-07029-f001:**
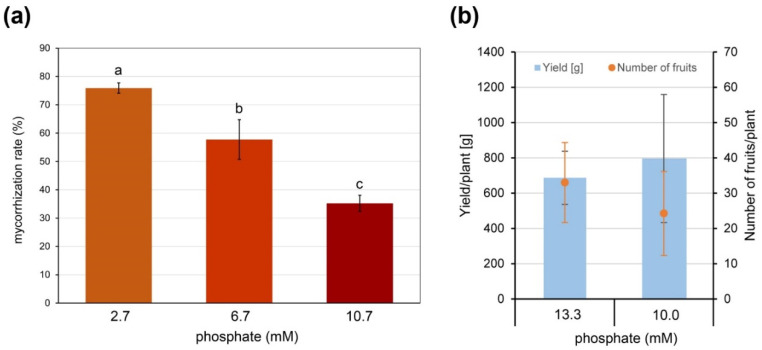
Effect of Pi-availability on formation of arbuscular mycorrhiza (AM) and yield. (**a**) Mycorrhization rate of roots of cv. Brioso inoculated with *Rhizophagus irregularis* and fertilized with 10 mL Long Ashton fertilizer containing 2.7, 6.7 or 10.7 mM Pi twice a week. Roots were harvested 17 weeks after inoculation. Shown are mean values ± SD (*n* = 3). Different letters indicate significant differences tested by 1-factorial ANOVA followed by Tukey HSD-test with *p* < 0.05. (**b**) Yield and number of fruits per plant of cv. Brioso under different fertilizer regimes. Plants were grown for 8 months and fertilized via drip irrigation using Long Ashton fertilizer containing 13.3 or 10.0 mM phosphate. Shown are mean values ± SD (*n* = 4).

**Figure 2 ijms-21-07029-f002:**
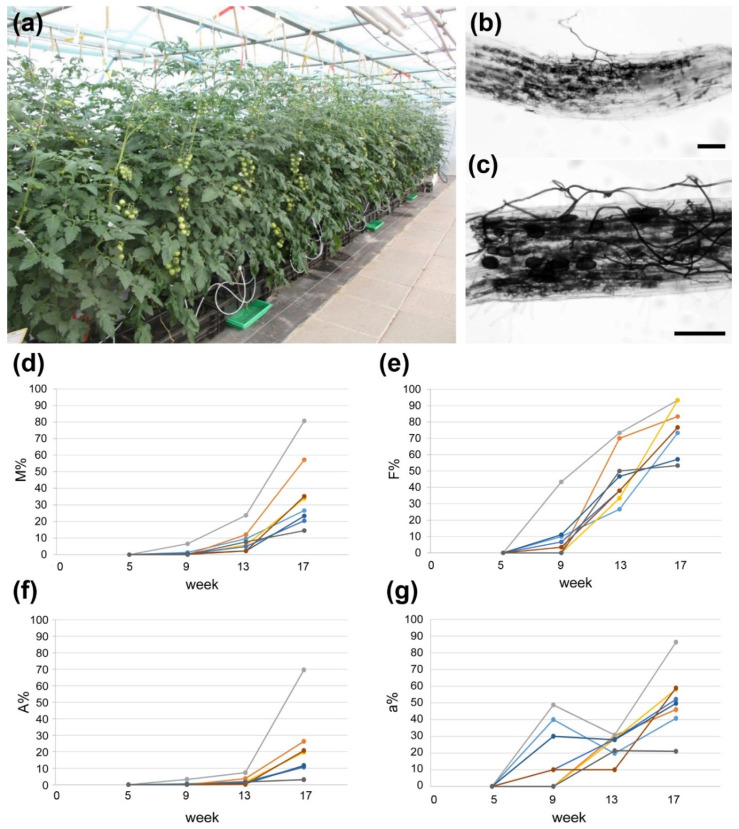
Mycorrhization of tomato plants (cv. Picolino/Maxifour) under producer-like conditions. (**a**) Cultivation set-up at INOQ GmbH (August 2019) using “Hawita special mix” as substrate and fertilization with 180 µM P_2_O_5_ (YaraTera/White fertilizer). (**b**,**c**) Ink-stained roots from mycorrhizal plants at the end of cultivation period showing fungal hyphae and arbuscules. Bars represent 100 µm. (**d**) Development of AM intensity (M%), (**e**) AM frequency (F%) and (**f**) arbuscule frequency (A%) in the root systems of eight single plants inoculated with *R. irregularis*. (**g**) Development of arbuscule abundance in mycorrhizal parts of root fragments (a%) of eight single plants inoculated with *R. irregularis*. For (**d**–**g**) time is given as weeks after inoculation.

**Figure 3 ijms-21-07029-f003:**
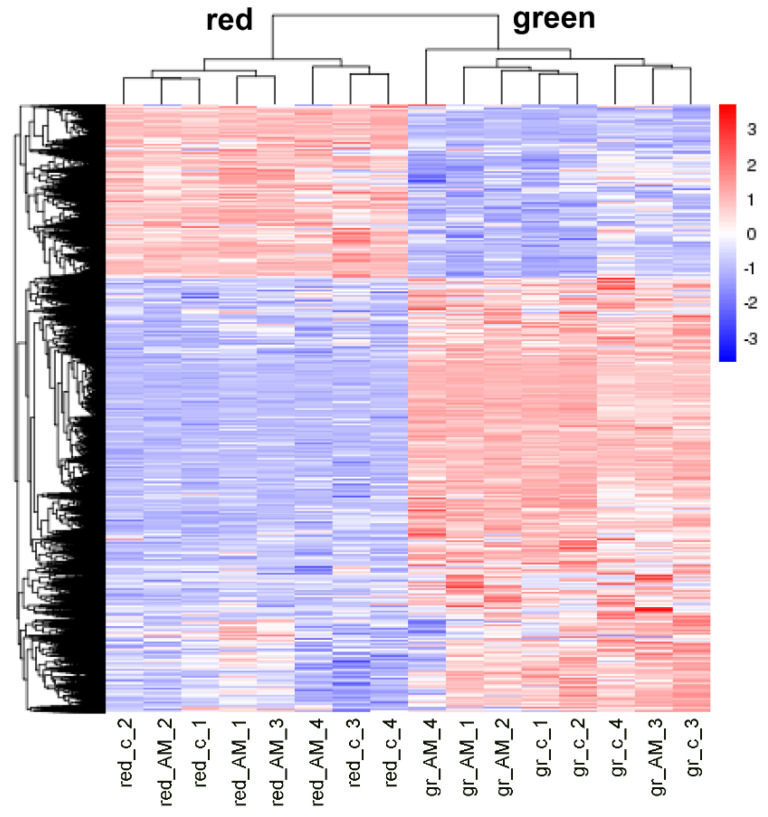
Heatmap showing hierarchal clustering of differentially expressed genes (DEGs). Clustering was performed using log10 (FPKM+1)-values of DEGs between green (gr) and red fruits of mycorrhizal (AM) and non-mycorrhizal (c) plants (*n* = 4, *p* < 0.05). Colors visualize column-scaled Z-score from low (blue) to high (red) gene expression.

**Figure 4 ijms-21-07029-f004:**
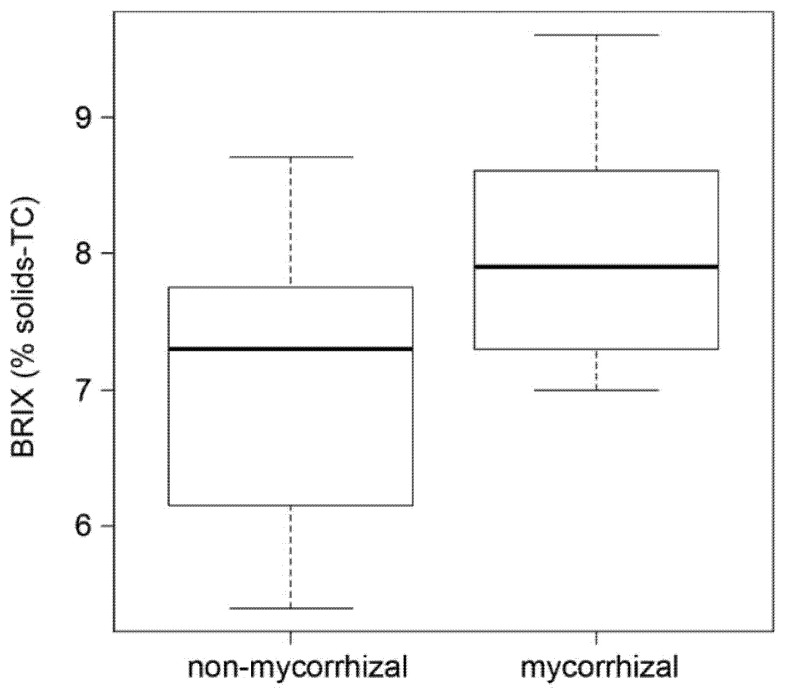
BRIX values of red fruits of non-mycorrhizal and mycorrhizal plants. BRIX was determined as a percentage of solids, temperature compensated (TC). Boxplots show medians (cross bar), 25–75% interquartile range (boxes) and data distribution (error bars). Students *t*-tests revealed no significant differences between both data sets (*p* = 0.08, *n* = 8).

**Figure 5 ijms-21-07029-f005:**
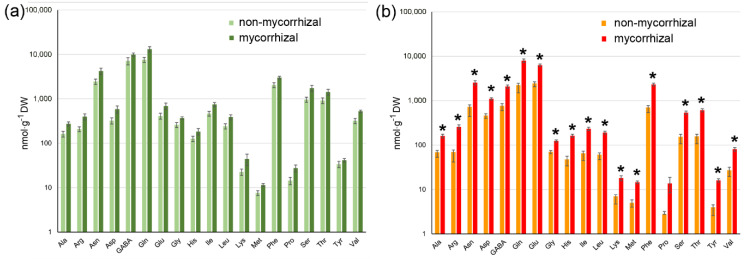
Levels of amino acids in green (**a**) and red (**b**) fruits of non-mycorrhizal and mycorrhizal tomato plants. Amino acids were determined by HPLC using methanol extracts from freeze-dried material. Note that there are no significant differences in the contents of any amino acids in green fruits, whereas all levels except that of proline were increased in red fruits upon mycorrhization of plants. Data are given as means ± SE (*n* = 8). * marks significant differences (*p* < 0.05) between fruits from non-mycorrhizal and mycorrhizal plants according to 1-factorial ANOVA including Bonferroni correction.

**Table 1 ijms-21-07029-t001:** Expression level of selected DEGs in green fruits of mycorrhizal (+AM) and non-mycorrhizal (−AM) plants.

Gene Name ^1^	Solyc No	−AM	+AM	Log2FC	−AM		+AM		Log2FC
		FPKM	FPKM		rEx ^2^	SE	rEx ^2^	SE	
*Dehydrin1*	*Solyc02g084840.3*	0.081	41.328	9.0	0.0019	0.0018	0.6899	0.6234	8.5
*malic enzyme*	*Solyc12g008430.2*	0.038	5.033	7.0	0.0371	0.0075	0.1357	0.1027	1.9
*2S albumin seed storage*	*Solyc07g064210.2*	0.138	16.656	6.9	0.0011	0.0011	0.1695	0.0890	7.2
*LEA 4*	*Solyc10g078780.2*	0.054	5.674	6.7	0.0008	0.0008	0.1121	0.1007	7.2
*LEA*	*Solyc02g062770.2*	0.127	11.423	6.5	0.0015	0.0015	0.2108	0.1842	7.2
*MADS-box TF 1*	*Solyc04g078300.3*	0.019	1.173	5.9	0.0000	0.0000	0.0147	0.0112	∞
*Aminotrans-ferase*	*Solyc01g007940.3*	0.051	2.819	5.8	0.0009	0.0009	0.0571	0.0389	6.0
*Vicilin*	*Solyc02g085590.3*	0.177	6.867	5.3	0.0059	0.0037	0.2590	0.1765	5.5
*bZIP TF*	*Solyc10g080410.2*	0.007	0.278	5.3	0.0000	0.0000	0.0112	0.0057	∞
*Zinc finger TF 50*	*Solyc07g053750.1*	0.139	4.944	5.2	0.0017	0.0007	0.0548	0.0402	5.0
*Desiccation-related*	*Solyc05g053350.3*	0.467	13.981	4.9	0.0060	0.0024	0.2046	0.1108	5.1
*PIN5*	*Solyc01g068410.3*	0.042	0.766	4.2	0.0004	0.0004	0.0076	0.0046	4.3
*ERF13*	*Solyc04g080910.1*	0.050	0.896	4.2	0.0002	0.0002	0.0130	0.0068	5.8
*Oleosin*	*Solyc06g069260.1*	0.162	2.715	4.1	0.0007	0.0007	0.0233	0.0109	5.0
*ACC-Oxidase5*	*Solyc07g026650.3*	10.087	4.424	−1.2	0.0271	0.0077	0.0108	0.0023	−1.3

^1^ DEGs were selected according to RNAseq analysis. RNAseq data are presented by fragments per kilobase of exon per million reads mapped (FPKM) values given as means (*n* = 4) and log2 fold changes (FC); ^2^ validation done using RT-qPCR analysis is shown as relative expression (rel. expr.) in relation to *SlTIP41* including the respective standard error (SE). Students *t*-test with *p* ≤ 0.05 done for RT-qPCR results showed no significant differences.

**Table 2 ijms-21-07029-t002:** Contents of selected carotenoids in fruits from non-mycorrhizal and mycorrhizal plants.

Carotenoid ^1^	Green Fruits		Red Fruits	
	−AM	+AM	−AM	+AM
Lutein	3.08 ± 0.46	3.44 ± 0.48	1.27 ± 0.41	1.22 ± 0.24
Zeaxanthin	4.36 ± 1.41	5.86 ± 1.19	n.d.	n.d.
Lycopene	n.d.	n.d.	2.87 ± 0.71	4.22 ± 1.97
ß-Carotene	0.63 ± 0.14	0.62 ± 0.19	7.10 ± 1.09	9.23 ± 2.15

^1^ Levels of carotenoids were determined by HPLC and are given as ng·mg^−1^ dry weight. Data are presented as means ± SD and did not show significant differences between fruits from non-mycorrhizal (−AM) and mycorrhizal (+AM) plants.

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
