# Peer review of "Effects of Arbuscular Mycorrhization on Fruit Quality in Industrialized Tomato Production"

_ijms, 2020, doi:10.3390/ijms21197029_

Round 1

Reviewer 1 Report

Schubert and co-workers analyses the effects of arbuscular mycorrhization on fruit quality in tomato production under commercial hydroponic system. Previous studies have revealed certain positive effects in tomato growth and development due to the mycorrhization. Mainly, mycorrhization increased fruit yield, shortens the vegetative growth period of tomato plants and leads to an earlier maturation of flowers and fruits. In addition, there is evidence that the mycorrhization increases the nutrient quality of fruits (increasing the content of citric acid, carotenoids and certain amino acids as well as the antioxidant capacity of the fruits) and has an impact on the transcriptome profile of these fruits.

In this study, the authors try to demonstrate these positive effects under soil-less hydroponic cultivation growth conditions suitable for tomato producers and Arbuscular Mycorrhiza (AM) formation in the greenhouse. Commercially used tomato cultivars (cv. Brioso and Picolino) were chosen in combination with the model AM fungal species Rhizophagus irregularis. Modifying substrate composition and Pi concentration and analysing changes in fruit gene expression and metabolite levels determination, the authors define the growing conditions in which AM enhances the nutritional quality of tomatoes without impairing plant growth and fruit yield. These conditions include a mix of substrate contains only low amounts of coco peat and a reduction of 20% in normal Pi concentrations.

Although the work is well planned and the experiments are well executed, study is recurrent and little innovative because is a repeat of previous trials under somewhat different experimental conditions. It is really difficult to find new data that provide knowledge on the subject. There are some points that must be clarified or discussed in greater depth.

General comments:

I have assumed (I have not found it explicitly in the text) that the normal Pi conditions in the commercial hydroponic system are 13.3 mM. The effect of reduced Pi (to 10 mM ) was tested in non-mycorrhizal plants and yield in terms of totally harvested tomatoes and number of tomatoes per plant was recorded and did not show significant differences. Further, yields from non-mycorrhizal (‐ AM) and mycorrhizal (+ AM) plants of tomato plants cv. Picolino grown under commercial conditions in greenhouse were not different. Does this mean that the concentration of Pi could be reduced to 80% without the need for mycorrhization and the same yields would be obtained????

As authors mentions, the use of the tomato cultivar as well as the fungal species may determine, whether increase in yield can be obtained. However, under soil conditions, AM fungal hyphae explore more territory and supply phosphorous and other nutrients to the plant. However, in controlled nutrition situations (soil-less hydroponic) the amount of Pi supplied is limited, as well as the exploration surface, and the hyphae cannot access an additional amount of the nutrients. Furthermore, nutrients are applied in forms very assimilable by the plant and must be shared by both organisms (micro and macro-symbiont). This reasoning leads me to ask if really the presence of mycorrhization in these culture conditions is a positive factor for plant growth and yield. The data provided suggests this. Do you agree with this argument? Is any nutritional effect to be expected in these conditions?

Another issue is the effects induced by mycorrhization and not directly related to the nutritional status of the plant. Within this group, the systemic effects on transcriptional changes in fruit or induced metabolic alterations could be found. Overall, the data indicated an only weak effect of arbuscular mycorrhization on the fruit transcriptome in tomato under greenhouse conditions and affects mainly the green fruits. Previous results of the global transcriptomic profile of the tomato fruit harvested at the turning stage of maturation revealed that at least twenty genes related to nitrogen and carbohydrate metabolism as well as in regulation and signal transduction showed a differential expression in the myc condition when compared to the control condition (Salvioli et al. 2012; Zouari et al. 2014).  Is there any similarity between the data shown in Table 1 and the data published in the previous study???

In the section concerning to AM effect on tomato fruit quality, only data from amino acids level seem unquestionable, since the data referring to carotenoids or BRIX values are less conclusive. Brix value alone is not a measurement or scale of fruit desirability. Brix measurements are perhaps most appropriate for measuring the degree of maturity rather than to compare fruit taste.

Red fruits from the mycorrhizal plants had significantly higher levels in all amino acids except proline. Higher contents of glutamine and asparagine have been also found in fruits of tomato plants interacting with F. mosseae, but only in the turning stage and not in the red stage (Salvioli et al., 2012). Any idea about why the same effect is expressed at different ripening time between both experiments???

Specific points:

Figure 1: Please specify the harvest times for each experiment

Figure 2: Data from part a, b and c does not contribute anything. it would be interesting to have data on the efficiency of the symbiosis (% arbuscules and / or expression of symbiotic genes)

It is not clear in the text the culture conditions for each treatment: 10.7 mM Pi in all conditions or 13,3 mM in control conditions ??

Figure 4: Are the NI vs. I data really significantly different? Differences were considered significant at a probability level of p < 0.05.

Reviewer 2 Report

The original research article titled ‘Effects of Arbuscular Mycorrhization on Fruit Quality in Industrialized Tomato Production’ submitted to International Journal of Molecular Sciences fits with the scope of the Journal. The authors conducted a very nice experiment on the interactive effect between arbuscular mycorrhizal (AM) fungi and phosphate concentrations on industrialized tomato production. Fruit quality features, changes in fruit gene expression and metabolite levels were checked by RNAseq and metabolite determination. From my point of view, the work is very well written and the originality is very high since to my knowledge this is the first report on the genes identification to highlight differences expressed between fruits from non-mycorrhizal and mycorrhizal plants in order to find markers indicating tomato quality improvement. The results of the paper are of high interest for tomato growers, nutritionists and scientists. The experimental design is very solid and the statistical analysis is of high quality.

I am really convinced that this paper meets the high standard of the Journal and will be highly cited.

However, I recommend to the authors to implement the Introduction section by citing the most recent articles on the use of AM fungi to improve fruit quality in Solanaceous crops such as the following:

  • Sabatino, L.; Iapichino, G.; Consentino, B.B.; D’Anna, F.; Rouphael, Y. Rootstock and Arbuscular Mycorrhiza Combinatorial Effects on Eggplant Crop Performance and Fruit Quality under Greenhouse Conditions. Agronomy 202010, 693.

Based on the above considerations I recommend the Editor to accept the manuscript for publication in International Journal of Molecular Sciences MDPI after the few minor revisions.

Author Response

Thanks the reviewer for the nice comments on our manuscript and the recommendation to cite the paper from Sabatino et al. We are sorry that we overlooked it. Now, we included the reference into the introduction (lines 65-66).

Reviewer 3 Report

I find the paper interesting, but I have some obstacles to assume it as a complete one:

  1. In a greenhouse experiment you have chosen grafted tomato plants ('Picolino' on 'MaxiFour'). There is no word on AMF-rootstock-scion relationships, and the effect of the graftage on the components, especially assuming the rootstock root system properties. It should be discussed.
  2. From the reason above, every time the material plant  should be described as cv. Picolino/MaxiFour.
  3. How could you explain the very high value of F% and M% parameters? It is rather not typical for: organic media, possible temporary root zone anoxia, and finally - easily accesible nutrients? 

Author Response

Reviewer:     In a greenhouse experiment you have chosen grafted tomato plants ('Picolino' on 'MaxiFour'). There is no word on AMF-rootstock-scion relationships, and the effect of the graftage on the components, especially assuming the rootstock root system properties. It should be discussed.

Answer: Thank you for this comment. We have introduced sentences about the graftage of tomato plants into the Results (lines 133-135) and the Discussion (lines 279-281).

Reviewer:    From the reason above, every time the material plant  should be described as cv. Picolino/MaxiFour.

Answer: According to the reviewer’s comment, the full description of the plant material has been introduced (lines 87, 141, 286, 305, 380, 407).

Reviewer:    How could you explain the very high value of F% and M% parameters? It is rather not typical for: organic media, possible temporary root zone anoxia, and finally - easily accessible nutrients?

Answer: The mycorrhization parameters varied a lot between the single plants analyzed (shown in Fig. 2e and f). It is true that the values are relatively high, but this might be due to the long growing period (17 weeks after inoculation). Moreover, even the highest values obtained are still lower than mycorrhization levels seen in plants grown under stronger phosphate limitation and laboratory conditions. We mention this now in lines 152-154: “. Due to the long growth period, however, the mycorrhization rates achieved are relatively high and reached levels also known from plants grown under stronger P-limitations and laboratory conditions (e.g., see [24]).”

Round 2

Reviewer 1 Report

The re-submitted version does not add anything new or substantially modify the work. In my opinion, the paper lack innovation and  it is difficult to find new data that provide knowledge on the subject.

Author Response

We very much regret that the reviewer finds our manuscript not containing sufficient new and substantial data. We are convinced, however, that we have been able to show for the first time that it is possible to get mycorrhizal tomato plants grown industrially in greenhouses. This was achieved by modifying substrate and fertilizer. In addition, we showed convincingly that even under these conditions mycorrhization results in an improvement in fruit quality as visible by a slightly altered fruit transcriptome and enhanced levels of amino acids in mature fruits. Since the other two reviewers and the editor consider our work worthy of publication, we are sticking to it.